# Structural Properties of Casein Micelles with Adjusted Micellar Calcium Phosphate Content

**DOI:** 10.3390/foods13020322

**Published:** 2024-01-19

**Authors:** Elaheh Ahmadi, Tatijana Markoska, Thom Huppertz, Todor Vasiljevic

**Affiliations:** 1Advanced Food Systems Research Unit, Institute for Sustainable Industries and Liveable Cities, College of Sport, Health and Engineering, Victoria University, Melbourne, VIC 3001, Australia; elaheh.ahmadi@live.vu.edu.au (E.A.); tatijana.markoska@live.vu.edu.au (T.M.); thom.huppertz@wur.nl (T.H.); 2FrieslandCampina, 3818 LE Amersfoort, The Netherlands; 3Food Quality and Design Group, Wageningen University and Research, 6708 WG Wageningen, The Netherlands

**Keywords:** micellar Ca phosphate, casein micelle, secondary structure of proteins, FTIR, NMR

## Abstract

Micellar calcium phosphate (MCP) content of skim milk was modified by pH adjustment followed by dialysis. Turbidity, casein micelle size and partitioning of Ca and caseins between the colloidal and soluble phases of milk were determined. Protein structure was characterised by Fourier transform infrared (FTIR) spectroscopy and proton nuclear magnetic resonance (^1^H NMR), whereas organic and inorganic phosphorus were studied by phosphorus-31 nuclear magnetic resonance (^31^P NMR). The sample with the lowest MCP content (MCP7) exhibited the smallest particle size and turbidity, measuring 83 ± 8 nm and 0.08 ± 0.01 cm^−1^, respectively. Concentrations of soluble caseins increased with decreasing MCP levels. At ~60% MCP removal, FTIR analysis indicated a critical stage of structural rearrangement and ^31^P NMR analysis showed an increase in signal intensity for Ca-free Ser-P, which further increased as MCP concentration was further reduced. In conclusion, this study highlighted the importance of MCP in maintaining micellar structure and its impact on the integrity of casein micelle.

## 1. Introduction

Milk is a highly nutritious food containing both macronutrients, such as proteins (caseins and whey proteins) and fats, as well as micronutrients, including minerals such as Ca. The capacity of caseins to bind calcium in milk enables them to transport calcium and phosphate at concentrations significantly exceeding the solubility of calcium phosphate. This is possible because the majority of Ca and inorganic phosphate (P_i_) in milk is in the form of micellar calcium phosphate (MCP) within the casein micelles. Casein micelles are heterogeneous, dynamic, polydisperse spherical structures with an average diameter of 200 nm [1]. They are composed of the four caseins, α_s1_-, α_s2_-, β-, and κ-casein, which are connected via various intermolecular interactions, as well as via MCP nanoclusters that connect to caseins via phosphoserines located on the former three caseins. The dry matter of the casein micelle consists of ~94% protein and ~6% inorganic material [2].

Of the total Ca content in bovine milk, ~70% is found in the casein micelles (protein-bound Ca, micellar) and ~30% is found in the serum phase (diffusible Ca) [3]. Serum Ca is partially present in ionic form, but also as complexes with citrate and phosphate ions. Conditions such as temperature, pH, as well as the addition of Ca-sequestering salts (CSS) can alter the amount of MCP, but also affect the Ca species in the serum phase. Alteration of the MCP level of the casein micelles can affect the structure and properties of casein micelles, which may further broaden the application range for milk protein ingredients in the food, cosmetic, and medicine domains [3].

The amount of soluble calcium affects rennetability, thermal stability, and rheological properties of various dairy products thus, calcium reduction was proposed as a way to improve the functionality of various dairy products. Due to the requirement for specific techno-functional properties of caseins as well as their nutritional benefits, in recent years, various studies have been conducted and different process options for calcium reduction in skim milk were compared quantitatively and qualitatively to produce casein micelles with an altered MCP content, e.g., in skim milk retentates [4] and in milk [5,6,7,8,9,10,11]. While functional properties of MCP-adjusted milks, such as thermal stability [12,13,14] and digestion [10] have been studied in detail, detailed studies on MCP-adjusted casein micelles have been more limited and the secondary structure of the proteins in MCP-adjusted skim milk was not investigated. This study aimed to investigate how modifying the MCP content in skim milk influences the structural characteristics of casein micelles, specifically focusing on its impact on micellar integrity.

## 2. Materials and Methods

### 2.1. Materials

Freshly pasteurised skim milk was sourced from Warrnambool Cheese and Butter—Saputo in Warrnambool, Victoria, Australia. Sodium azide (0.02%, *w*/*w*), glucono-delta-lactone (GDL), sodium hydroxide (NaOH), deuterium oxide (D_2_O) and Pronase (protease mixture) were procured from Sigma-Aldrich in St. Louis, MO, USA. Additionally, high retention seamless cellulose dialysis tubing (14 kDa MWCO) was obtained from Sigma-Aldrich, St. Louis, MO, USA.

### 2.2. Sample Preparation

Samples were prepared from freshly pasteurised skim milk obtained from Warrnambool Cheese and Butter—Saputo (Warrnambool, Victoria, Australia) on three separate occasions. To prevent bacterial growth, sodium azide (0.02%, *w*/*w*) was added to the milk on receipt. In order to prepare samples varying in MCP content, the skim milk was subjected to a protocol described previously [5], whereby pH of skim milk was first either lowered to 4.9, 5.5, 5.7, 5.8, 5.9 or 6.1 by the addition of predetermined amounts of glucono-delta-lactone (GDL) or increased to 7.5 or 8.2 by adding 1.0 M NaOH at 5 °C. After the pH was stabilised, the pH-adjusted samples, as well as a control sample maintained at pH 6.7, were dialysed using a high retention seamless cellulose dialysis tubing (14 kDa MWCO, Sigma-Aldrich, St. Louis, MO, USA), against 2 × 20 volumes of original pasteurised skim milk for 72 h at 5 °C [5,10]. After the dialysis, the samples were removed from dialysis tubing and analysed. The study design of the present study and associated sample coding is shown in Figure 1. The coding was based on the estimate of micellar Ca relative to that of the control. Micellar Ca was defined as the amount of Ca that sedimented on ultracentrifugation (Section 2.3).

### 2.3. Sample Fractionation

The sedimentable and non-sedimentable phase of milk samples were separated by ultracentrifugation at 100,000× *g* for 1 h at 20 °C using a Beckman Ultra L-70 centrifuge (Beckman Coulter, Australia Pty. Ltd., Gladesville, Australia). The clear supernatant was then collected carefully with a syringe from each tube. To prepare 10 kDa-permeable fractions, part of the ultracentrifugal supernatant was subsequently filtered through a centrifugal 10 kDa filter at 4000× *g* for 4 h (Corning Spin-X UF concentrators, Merck, Darmstadt, Germany) using an Eppendorf Model 5810 centrifuge (Hamburg, Germany). A 10 kDa-permeable fraction was also prepared, similar to outlined above, from milk samples which had been incubated with 0.4 mg mL^−1^ of the protease mixture Pronase from *Streptomyces griseus* type XIV (Sigma-Aldrich, ST. Louis, MO, USA) for 24 h at 20 °C.

### 2.4. Sample Analysis

#### 2.4.1. Ca Determination

The concentration of Ca in the whole samples, ultracentrifugal supernatant, and the 10 kDa permeates of milk with and without prior pre-treatment with Pronase was determined using an inductively coupled plasma atomic emission spectrometer (ICP-AES, ICPE-9000 system, Shimadzu Corporation, Kyoto, Japan) [15]. Micellar Ca was defined as the amount of sedimentable Ca (i.e., total Ca–Ca in the ultracentrifugal supernatant). The amount of nanocluster-associated Ca was determined as the difference between total Ca and the amount of Ca in the 10 kDa permeate of Pronase-treated milk.

#### 2.4.2. Particle Size Distribution and Turbidity

Particle size (Z-average diameter) was determined by dynamic light scattering (Zetasizer-Nano, Malvern instruments Ltd., Malvern, UK) at a scattering angle of 90° at 25° C. Samples were diluted in simulated milk ultrafiltrate (SMUF) [16] in a ratio of 1:100. Refractive indexes for casein micelle and SMUF of 1.57 and 1.342, respectively, were used [17]. Turbidity of the bulk samples was measured at 860 nm using 1 mm path length quartz cuvettes using a UV–Visible spectrophotometer (Biochrom Ltd., Cambridge, UK) following the previously described method [15]. Turbidity of the milk serum after centrifugation was also measured.

#### 2.4.3. High-Performance Liquid Chromatography

Caseins in whole samples and ultracentrifugal supernatants were analysed using a reversed-phase high-performance liquid chromatography (RP-HPLC) at room temperature by a Shimadzu HPLC system (Model Prominence-i, LC- 2030 C, Shimadzu Corporation, Kyoto, Japan) with a Varian 9012 system controller (Agilent Technologies Inc., Santa Clara, CA, USA) coupled with a RI detector (Varian, 9050) and a C_4_ column (Aeris Widepore, 150 mm × 4.6 mm, 3.6 μm particle size, 300 Å porosity, Phenomenex, Torrance, CA, USA). Samples were prepared and analysed according to the method described previously [15].

#### 2.4.4. Fourier Transform Infrared (FTIR) Spectroscopy

For FTIR spectroscopy measurements, all milk samples were analysed using an FTIR spectrometer (Frontier 1, PerkinElmer, Boston, MA, USA) in the range of 4000–600 cm^−1^. At the start of measurement, the background spectrum was scanned with a blank (SMUF) to resolve changes in milk proteins [18]. In addition, corresponding ultracentrifugal supernatant samples were analysed under the same instrumental conditions as for the milk sample spectra acquisition [18]. The spectrum of ultracentrifugal supernatants was subtracted from corresponding spectrum of milk samples to reveal structural features of the micellar casein fraction. In the Amide I band region of the FTIR spectra between 1700 and 1600 cm^−1^, six features corresponding to main protein secondary structures were assigned: side chains (1608–1611 cm^−1^), β-sheet (1620–1631 cm^−1^), random coils (1640–1649 cm^−1^), α-helix (1658–1666 cm^−1^), β-turns (1668–1681 cm^−1^), and aggregated β-sheet (1689–1694 cm^−1^) [19]. By subtracting the supernatants from the corresponding bulk samples, the aim was to visualise the changes associated with the micellar casein phase.

#### 2.4.5. NMR Spectroscopy

NMR analysis was carried out using a 600 MHz Bruker Avance spectrometer (Bruker BioSpin GmbH, Rheinstetten, Germany). For the ^31^P NMR analysis, the samples were prepared by mixing 0.1 mL of deuterium oxide and 0.9 mL of milk sample, whereas for the ^1^H NMR analysis the samples were mixed by 0.9 mL of deuterium oxide 10% and 0.1 mL of milk. The ^1^H NMR spectra were recorded using 32 scans and spectral width of 9615 Hz. The water signal was suppressed using excitation sculpting with gradients allowing for presaturation during relaxation delay in cases of radiation damping [19]. The ^31^P NMR spectra were acquired at frequency of 242 MHz with power-gated proton decoupling, acquisition time of 0.3 s and 36 number of scans. The spectra were analysed using TopSpin 4.1.1 software (Bruker BioSpin). The FID was corrected by 0.3 Hz line-broadening parameter and phase correction by 0th and 1st order correction for pk. For both analyses, ^1^H and ^31^P NMR, each sample was analysed in triplicate.

### 2.5. Statistical Analysis

All experiments assessed the impact of MCP content on the selected parameters. SPSS software v. 26 (IBM Inc. Chicago, IL, USA) was used for one-way analysis of variance (ANOVA) to establish differences among means followed by Tukey’s multicomparison of the means. The level of significance was pre-set at *p* < 0.05. The data were replicated three times on three different occasions. In addition, the FTIR data, processed as described previously, was analysed by Principal Component Analysis (PCA) with Origin Pro 2021, v. 95E software (OriginLab Corporation, Northampton, MA, USA), as described previously [18].

## 3. Results

### 3.1. Ca Distribution

Table 1 shows the concentrations of Ca in the different fractions of MCP-adjusted skim milk samples. The total Ca content of the control skim milk (MCP_100_) was ∼32.2 mmol L^−1^, out of which 21.5 mmol L^−1^ was colloidal, 10.7 mmol L^−1^ was found in the ultracentrifugal supernatant and 9.7 mmol L^−1^ was 10 kDa permeable. Concentrations of Ca in the ultracentrifugal supernatants were slightly higher than the corresponding levels in the 10 kDa-permeable fractions, due to fact that whey proteins and caseins in the ultracentrifugal supernatant bind some Ca. In agreement with previous studies [5,10], MCP levels decreased in samples that were dialysed after acidification, whereas it increased in samples that were alkalinised prior to dialysis (Table 1). In addition to micellar Ca, we also estimated nanocluster-associated Ca, as previous studies indicated that not all micellar Ca is in the MCP nanoclusters [20]. To do this, the level of Ca in the 10 kDa-permeable fraction of milk that treated with Pronase was determined.

Treatment of milk with this enzyme mixture is known to hydrolyse milk proteins to free amino acids and small peptides, which can permeate through a 10 kDa membrane, whereas the Ca phosphate nanoclusters remain intact [21,22]. Hence, Ca associated with other parts of the caseins would permeate through the 10 kDa membrane after hydrolysis with Pronase. Using this approach, the concentration of nanocluster-associated Ca was estimated to be 20.5 mmol L^−1^ in the control milk. With decreasing MCP content, the concentration of nanocluster-associated Ca decreased significantly, down to 2 mmol L^−1^ of the nanocluster associated Ca for sample MCP_7_ (Table 1).

### 3.2. Physicochemical Properties of Skim Milk with the Altered MCP Content

Particle size and turbidity of all samples are presented in the Table 2, whereas the particle size distribution of the milk samples is shown in Figure 2. The average particle size in control skim milk was 163 nm (Table 2), in line with previous reports [10]. With decreasing MCP content, the turbidity of skim milk samples decreased progressively, with the turbidity of the sample MCP_7_ close to that of the milk serum (Table 2), indicating a very high degree of disintegration of the casein micelle. On the other hand, increasing the MCP content did not alter the turbidity (Table 2).

### 3.3. Distribution of Individual Caseins in the MCP-Adjusted Skim Milk

Results for the protein composition of the ultracentrifugal supernatants of the MCP-adjusted skim milk samples as determined by HPLC are shown in Figure 3. The percentage of non-sedimentable α_S1_- α_S2_-, κ-, and β-casein increased with decreasing MCP content (Figure 3), particularly from sample MCP_67_. The highest concentrations of non-sedimentable caseins were found in sample MCP_7_. Increasing the MCP content caused slight decrease in the concentrations of individual soluble caseins in comparison to the control (MCP_100_) (Figure 3). Comparable trends were observed for all caseins.

### 3.4. Structural Characterisation of MCP Adjusted Skim Milk Samples by FTIR

The structural changes in the Amide I region of milk proteins and the micellar casein fraction in the samples are shown in Appendix A and Table 3, respectively. FTIR spectra in the region between 4000 and 650 cm^−1^ are presented in Appendix A. The most substantial difference as a result of adjusting the MCP content of skim milk was observed in the side chains of the amino acids (1608–1611 cm^−1^), with the greatest intensity observed for samples MCP_113_ and MCP_129_ (Appendix A; Table 3). The intensity of side chains in the micellar casein fraction decreased significantly at two key points when MCP content was reduced (Table 3). The first notable decrease was observed between samples MCP_100_ and MCP_67_, and the second significant change was observed between samples MCP_58_ and MCP_42_ (Table 3). It is important to note that the side chains of amino acids can contribute to the structural integrity of proteins through weak interactions, such as ionic or hydrogen bonds [23].

The greatest structural changes could be assigned to the micellar casein fraction. The proportion of side chains, which contribute to the structural integrity of proteins through weak interactions, such as ionic or hydrogen bonds [23], was greatest at the MCP level above that of the control, it however consistently declined concomitant with the reduction in the MCP content. On the contrary, the proportion of β-turns followed an inversed pattern to that of the side chains with the lowest values obtained at high MCP content. Random coil and β-sheet structures, with some exceptions, remained fairly consistent across the whole MCP range (Table 3). α-Helical structures significantly increased when lowering MCP content down to 42% of the original level, after which it declined by almost 30% (Table 3). At the same time, contributions of the β-sheet and aggregated β-sheet structures were the lowest for MCP_42_.

From these observations, sample MCP_42_ appeared to be a key point at which the structural components of the proteins underwent substantial changes as a result of changes in the MCP content. PCA analysis also confirmed differences in structural components in Amide I region based on MCP-adjustment, classifying the samples into three groups (Figure 4). The PC1 differentiated the MCP adjusted skim milk samples in three groups including the MCP-enriched samples (MCP_113_ and MCP_129_) from the low-MCP samples (MCP_7—_MCP_58_), and samples MCP_67_—MCP_100_ (Figure 4).

#### H and ^31^P NMR

The MCP-adjusted milk samples were analysed by ^31^P and ^1^H NMR spectroscopy. The ^31^P NMR spectra provide information on the changes in the state of phosphate in milk and the spectra presented in Figure 5 are characterised by two distinct peaks, one narrow peak at δ = 1–1.5 ppm assigned to P_i_ and a broader peak at around 3 ppm usually assigned to P_o_. In the control samples, the P_o_ signal at 2.0–3.5 ppm was very broad and started to become more prominent at MCP_42_ (Figure 5D) with the greatest intensity detected at MCP_7_ (Figure 5A). The signal shape and intensity of this peak depends on the different chemical environments and mobility of SerP [24], with the signal disappearing concomitant with the phosphorus immobilisation. The Po signal appeared to be noticeable different for the samples containing the least amount of MCP (Figure 5A–C), whereas the Pi peak appeared consistently across these ranges of MCP. The width of the Pi peak is impacted by the MCP concentration; in samples with higher MCP content (Figure 5H,I), a broader width is observed.

The ^1^H NMR spectra showed an intense signal for lactose at 3.2–4.0 ppm (Figure 6) [25]. Signal intensity in the aliphatic region (0.5–2.0 ppm) decreased as the MCP concentration was increased. The methyl signal that appears at ~0.5 ppm was most intense in MCP_7_ and the least in MCP_129_ (Figure 6). This resonance arises from the methyl protons of alanine, leucine, isoleucine and threonine and their intensity could be related to their greater presence in the soluble phase [26]. Similarly, the aromatic region of the spectrum (6–9 ppm) [27], depicting ring protons of phenylalanine, tyrosine, tryptophan and histidine, resonated greatly at lower MCP content (Figure 6A–E), which again was attributed to a greater proportion of soluble proteins in the past [28]. The Hα and Hβ regions of the ^1^H NMR spectra also showed variation in the signal intensity when the MCP concentration was altered (Figure 6). Changes were observed in the doublet at 2.5 ppm, doublet at 3.0 ppm, singlet at 4.2 ppm, singlet at 5.2 ppm and multiple signals at 4.7 ppm. The chemical shifts in the MCP_31_ sample at 2.5, 3.0, 4.2 and 5.2 ppm show lower signal intensity compared to the rest of the samples. Thus, the signal in this region could be due to involvement in hydrogen interactions guided by the Hα and Hβ of the backbone. The signals at 4.7 ppm could not be interpreted accurately as the water suppression interfered with the signal intensity.

## 4. Discussion

### 4.1. Impact of Micellar Calcium Phosphate Levels on the Casein Micelle Structure

Casein micelles are in a dynamic equilibrium with the surrounding serum; thus, they can exchange proteins and minerals under various saturation conditions, and rearrange their structure in different chemical environments [29]. De Kruif et al. [30] showed that the previously proposed nanocluster model [31] captures the main features of the casein micelle. In this model, the MCP is dispersed by competent phosphopeptides to form equilibrium core-shell nanoclusters [20]. Based on this model, phosphorylated caseins bind to the growing nanoclusters [30]. The proteins associated with the nanoclusters protrude out and interact with other proteins via weak interactions, including hydrophobic interactions, hydrogen bonding, ion bonding and weak electrostatic interactions, which results in formation of a more or less homogeneous protein matrix [30]. Previously, Holt et al. [32] stated that micellar Ca appears to exist in two forms—as a Ca phosphate salt and as Ca^2+^ bound to the protein. Similarly, originally it was believed that phosphorus appeared in 2 chemical forms in casein micelle: as the MCP and phosphorylated serine (phosphoserine) in the caseins [33,34]. However, more recently Hindmarsh and Watkinson showed using ^1^H ^31^P cross-polarisation magic angle spinning (CP-MAS) NMR that in addition to these two forms, other immobile phosphorus bodies exist within the casein micelle that have not yet been classified [35].

### 4.2. Micellar Calcium Phosphate Adjustment: Insights from FTIR Analysis

The state of MCP has a notable effect on the properties of the casein micelle. The results of the current study confirmed previous findings [8,9,10,15] that the MCP adjustment changes Ca equilibrium (Table 1), resulting in a substantial decline of the average particle size below MCP_42_. Interestingly, from the data in Table 2 it seems that turbidity declined almost linearly with decreasing MCP content. The caseins and MCP are responsible for the light-scattering properties of the casein micelle [36] thus decreasing MCP concentration reduces the refractive index of the casein micelles, and thus turbidity of the milk. However, a decrease in turbidity was also associated with a release of individual caseins [2], which indicated changes in the concentration of non-sedimentable caseins in the serum phase (Figure 3). β-Casein appeared to be the most affected micellar protein initially, with almost 40% of its initial micellar concentration released into the serum (Figure 3). This could be related to its positioning in the casein micelle, as some studies indicated that β-casein may either be present or be close to the surface [37,38]. The role of MCP in maintaining β-casein association with the micelle was also linked to its greater solubility at low temperature [39].

Structurally, the main changes observed were associated with the micellar and the non-sedimentable caseins. Only limited changes in structural elements took place when all milk proteins were assessed involving side chains, β-sheets and random coils while α-helix, β-turns and aggregated β-sheets remained fairly consistent (Appendix A). Changes were likely influenced by dissociation of individual caseins from the micelle, as whey proteins likely remain unaffected by the whole process of MCP adjustment. On the other hand, the casein micelle underwent substantial structural changes (Table 3), with changes becoming very prominent around residual MCP content of 42 to 58% of the original. The Amide I signal of the casein micelle was previously reported to contain ~40% of β-turns at the natural pH of milk [40], similar to what was observed in the current study (Table 3). The substantial increase of β-turns upon the MCP reduction could in part be attributed to dissolution of β-casein from the micelle since it contains approximately 20% of β-turns [40,41]. In addition, β-casein contains a large proportion of β-sheets [41] thus the decline in this structural feature of the casein micelle could have been caused also by the β-casein solubilisation. Considering other physical properties, it is likely that the structure of the casein micelle remains fairly stable up to a certain point, around MCP_42_, although the protein and mineral composition may have changed. This could be related to the proposed structure of MCP and how it interacts with the phosphoserine residues of the individual caseins.

### 4.3. Micellar Calcium Phosphate Adjustment: Insights from NMR Analysis

As established by ^31^P NMR, milk contains about 23 mM of inorganic phosphate (P_i_) and approximately 10 mM-phosphate esters, mainly as SerP residues of the caseins [42]. These phosphate assignments are clearly depicted as δ ~ +1.9 originating from P_i_ with, in general, a broad peak at approximately δ = +3.2 arising from SerP of the casein (Figure 5) [42]. Depending on the type of ^31^P NMR analysis, four types of phosphorus have been identified in literature including organic phosphorus from phosphoserine residues, organic phosphorus from serine to the MCP, inorganic phosphorus in the MCP and free inorganic phosphorus in the serum [43]. The MCP adjustment in our study appears to initially affect the width of the P_i_ peak (Figure 5) as it was wider at a higher MCP concentrations and then became narrower as the concentration of the MCP declined. This could be related to the types of P_i_ present in the MCP. Van Dijk [44] suggested that the MCP consisted of two parts—the stable arm, which crosslinks with SerP, and the unstable arm, which is in equilibrium with soluble salts in milk serum. Kolar et al. [45] extended on this suggestion and proposed that P_i_ appeared as three types including free inorganic phosphate in the serum, inorganic phosphate belonging to the unstable arm of the MCP and in the equilibrium with the serum P_i_, and inorganic phosphate as a part of the stable arm of the MCP. Such a model has been termed C_2_-SerP_3_ ion cluster and depicts 2 ‘free’ unstable and 2 stable arms firmly connecting 2 peptide chains [45]. It is likely that partial dissolution of the MCP would lead to solubilisation of the unstable MCP arms, which would not affect the integrity of the micelle greatly. However, once the stable arm is dissolved and SerP is exposed the micellar integrity appears to be compromised, which likely started to occur at MCP_42_ (Figure 5D). At this point, P_o_ started to appear, indicated a greater mobility of SerP (Figure 5D). The peak at +3.2 ppm became the most prominent at the lowest MCP content. This is also confirmed by ^1^H-NMR (Figure 6) as the peaks in 2 identified regions (0.5–3.5 ppm and 5.5–9.0 ppm) started to change. The resonance around 1 ppm, which arises from the methyl protons of Ala, Leu, Ile and Thr residues, appeared to increase concomitant with the MCP decrease. At the same time, the peaks in the 5.5–9.0 ppm region, derived from amide main and side chain and aromatic side chain protons from caseins, started to become more distinct at MCP_42–58_ (Figure 6D,E). The spectrum in the 7.5 and 8.5 ppm region has been assigned to the casein micelle backbone previously [46] and the peaks in this region were very broad but started to appear at MCP below 67% (Figure 6F). All of this indicates that either parts of the casein micelle became more mobile (resonated more) or the micelle lost its integrity, which resulted in greater resonance.

Increasing the MCP content by 13 or 29% had a rather minimal impact on the observed physical properties. However, it appears that the individual caseins are drawn from the soluble phase into the micellar structure. This resulted in the further immobilisation of organic phosphorus and widening of the P_i_ peak (Figure 5H,I), decline in the resonance associated with free amino acids depicted by ^1^H-NMR (Figure 6H,I) and decline in the concentration of non-sedimentable caseins (Figure 3). While it may be plausible that additional P_i_ could be incorporated into the unstable arm of the MCP, it is not clear how the individual caseins would be further incorporated into the casein micelle. This is especially in reference that the particle size did not change significantly thus one of the assumptions could be that they would be positioned somewhere in the interior. It is however evident that even increasing the MCP content above its native level would cause structural changes in the casein micelle including rise in side chains and aggregated β-sheets and decline in the α-helical and β-turn structures (Table 3). How these structural changes may impact the properties and behaviour of the casein micelle under various processing conditions remains to be determined.

## 5. Conclusions

Several important physicochemical changes in milk proteins take place when the content of MCP in the casein micelle is altered. The casein micelle integrity did not appear substantially affected by the decline in MCP concentration following its adjustment down to 42% of its original level, although the level of individual caseins increased in the serum phase of milk. The MCP content of 42% of the initial appears to be a minimum level to maintain the integrity of the casein micelle. While physical changes were not considerably noticeable above this level, the conformational changes in the casein micelle were substantially impacted by the adjustment. The MCP levels were also increased by adjustment at elevated pH although there appeared to be a limit to which additional Ca can be incorporated into the nanoclusters. This study provided some insights into the conformational and physicochemical changes in pasteurised milk with adjusted MCP.

## Figures and Tables

**Figure 1 foods-13-00322-f001:**
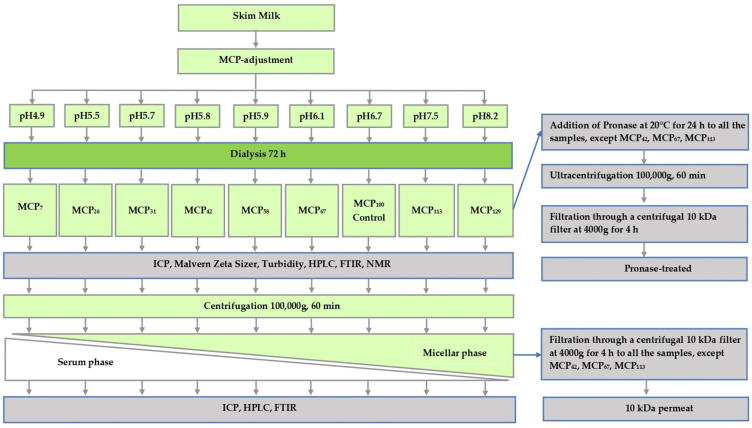
Experimental design of this study.

**Figure 2 foods-13-00322-f002:**
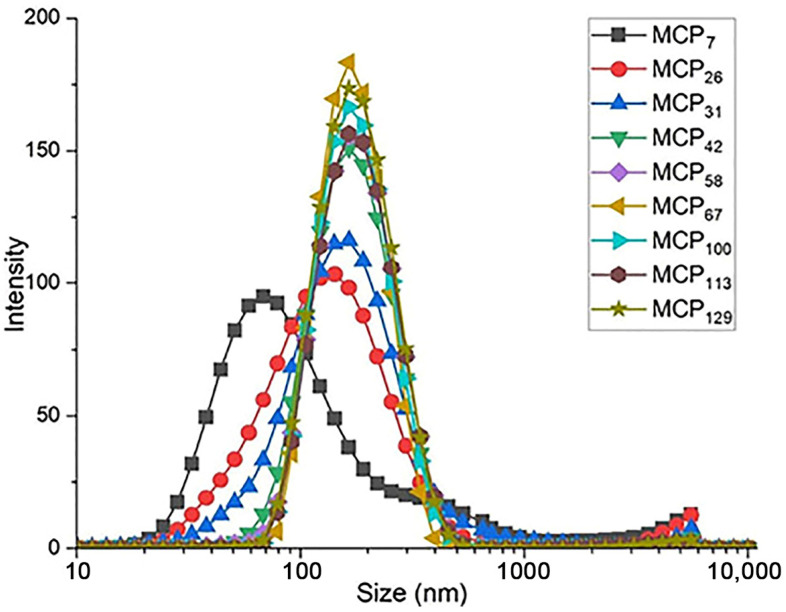
Particle size distribution of MCP modified skim milk samples. MCP content was adjusted from 7% (MCP_7_) to 129% (MCP_129_) relative to the control by either acidification or alkalisation followed by exhaustive dialysis against bulk milk. Graph is representative of two replicate samples.

**Figure 3 foods-13-00322-f003:**
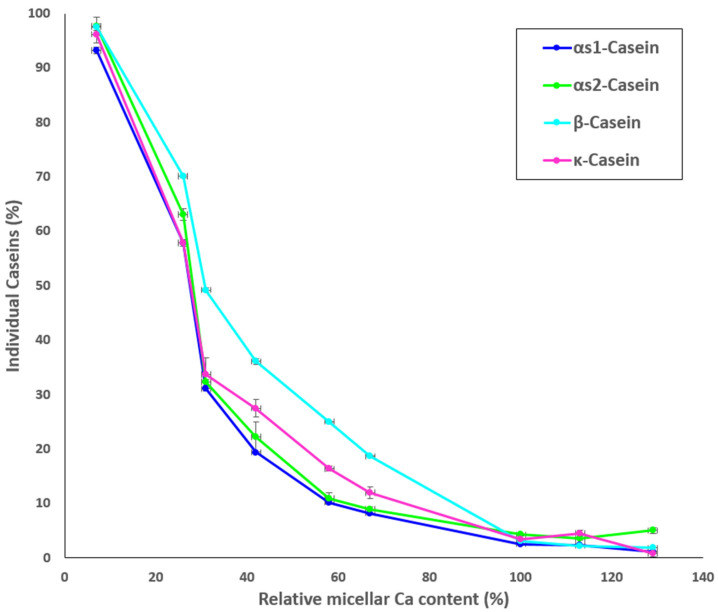
Proportion (%) of individual caseins present in the supernatant to skim milk as a function of the relative micellar Ca content, calculated as a ratio of micellar Ca content of a sample after pH adjustment and dialysis to micellar Ca content of the control.

**Figure 4 foods-13-00322-f004:**
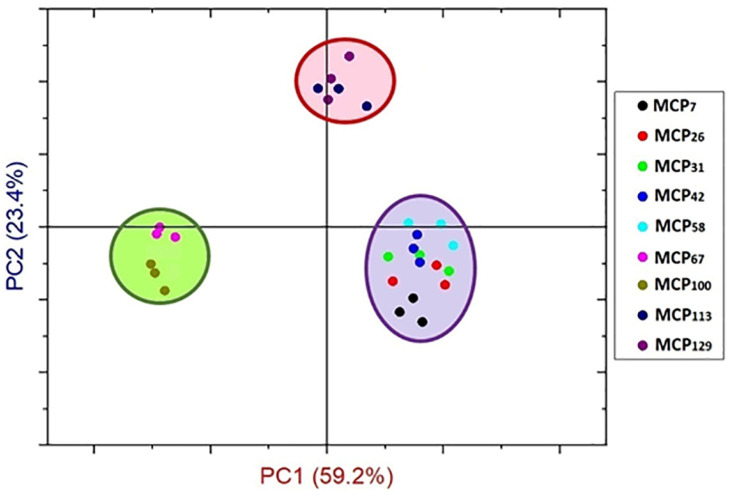
Principal component scores for Amide Ɪ region (1700–1600 cm^−1^) of MCP-adjusted skim milk samples containing from 7% (MCP_7_) to 129% (MCP_129_) of MCP relative to that of the control achieved by either acidification or alkalisation followed by exhaustive dialysis against bulk milk.

**Figure 5 foods-13-00322-f005:**
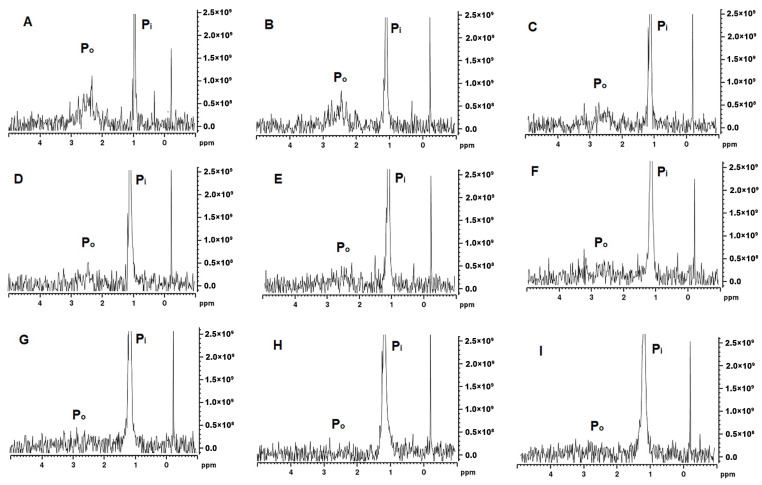
^31^P NMR spectra for MCP_7_ (**A**), MCP_26_ (**B**), MCP_31_ (**C**), MCP_42_ (**D**), MCP_58_ (**E**), MCP_67_ (**F**), MCP_100_ (**G**), MCP_113_ (**H**), and MCP_129_ (**I**).

**Figure 6 foods-13-00322-f006:**
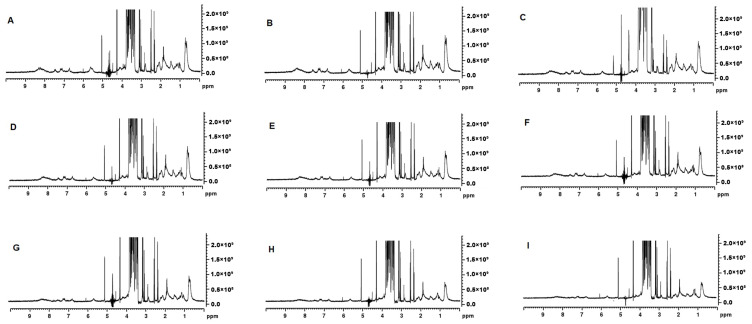
^1^H NMR spectra for MCP_7_ (**A**), MCP_26_ (**B**), MCP_31_ (**C**), MCP_42_ (**D**), MCP_58_ (**E**), MCP_67_ (**F**), MCP_100_ (**G**), MCP_113_ (**H**), and MCP_129_ (**I**).

**Table 1 foods-13-00322-t001:** Ca concentration in milk, its supernatant (100,000× *g* for 60 min), 10 kDa permeate and the 10 kDa permeate of milk after treatment with Pronase for pasteurised skim milk samples with their MCP adjusted to 7% (MCP_7_) to 129% (MCP_129_) by either acidification or alkalisation followed by exhaustive dialysis against bulk milk. MCP content was relative to the control based on the nanocluster associated Ca.

Sample Code ^1^	pH of Milk Prior to Dialysis	Total Ca (mmol L^−1^)	Supernatant Ca (mmol L^−1^)	10 kDa Permeable Ca (mmol L^−1^)	10 kDa-Permeable Ca of Pronase-Treated Milk (mmol L^−1^)	Nanocluster-Associated Ca (mmol L^−1^)
MCP_7_	4.9	14.74 ± 1.01 ^E^	13.24 ± 0.50 ^A^	12.49 ± 0.26 ^A^	12.65± 1.56	2.09 ± 0.06 ^F^
MCP_26_	5.5	18.24 ± 3.19 ^DE^	12.53 ± 2.19 ^AB^	11.48 ± 0.94 ^AB^	13.54 ± 1.38	4.70 ± 0.20 ^E^
MCP_31_	5.7	19.23 ± 5.06 ^DE^	12.51 ± 2.30 ^AB^	11.64± 0.53 ^AB^	12.84 ± 1.33	6.39 ± 0.09 ^D^
MCP_42_	5.8	21.38 ± 3.42 ^CD^	12.32 ± 2.07 ^AB^	N/A	N/A	N/A
MCP_58_	5.9	23.68 ± 3.38 ^CD^	11.08 ± 0.72 ^AB^	11.49 ± 2.38 ^AB^	13.21± 1.14	10.47 ± 0.06 ^C^
MCP_67_	6.1	25.42 ± 1.55 ^C^	10.92 ± 0.87 ^AB^	N/A	N/A	N/A
MCP_100_	6.7	32.23 ± 1.79 ^B^	10.72 ± 0.67 ^AB^	9.78 ± 0.09 ^B^	11.72± 2.33	20.51 ± 0.02 ^B^
MCP_113_	7.5	35.43 ± 2.83 ^AB^	11.04 ± 0.48 ^AB^	N/A	N/A	N/A
MCP_129_	8.2	38.17 ± 3.19 ^A^	10.47 ± 0.28 ^B^	11.63 ± 1.58 ^AB^	13.92± 1.57	24.25 ± 0.10 ^A^

^1^ The subscript numbers indicate proportion of retained MCP relative to that of the control; The superscript capital letters indicate significant differences within the rows across treatment by Tukey’s honestly significant difference procedure (*p* < 0.05); results are expressed as the means ± standard deviation; N/A—not assessed.

**Table 2 foods-13-00322-t002:** Particle diameter and turbidity for pasteurised skim milk samples with their MCP adjusted from 7% (MCP_7_) to 129% (MCP_129_) by either acidification or alkalisation followed by exhaustive dialysis against bulk milk For sample details, see Figure 1.

Sample	Particle Diameter (nm)	Turbidity (cm^−1^)
MCP_7_	83 ± 8 ^D^	0.08 ± 0.01 ^I^
MCP_26_	115 ± 7 ^C^	0.14 ± 0.01 ^H^
MCP_31_	140 ± 17 ^B^	0.23 ± 0.00 ^G^
MCP_42_	158 ± 14 ^A^	0.26 ± 0.00 ^F^
MCP_58_	163 ± 8 ^A^	0.29 ± 0.00 ^E^
MCP_67_	162 ± 5 ^A^	0.32 ± 0.00 ^D^
MCP_100_ (control)	163 ± 4 ^A^	0.41 ± 0.00 ^A^
MCP_113_	169 ± 11 ^A^	0.40 ± 0.00 ^B^
MCP_129_	165 ± 5 ^A^	0.39 ± 0.00 ^C^

The capital letters indicate significant differences within the rows across treatment by Tukey’s honestly significant difference procedure (*p* < 0.05); results are expressed as the means ± standard deviation.

**Table 3 foods-13-00322-t003:** Changes in structural features of individual caseins in Amide I region of MCP-adjusted skim milk samples as determined by Fourier transform Infrared Spectroscopy. MCP-adjusted skim milk samples containing from 7% (MCP_7_) to 129% (MCP_129_) of MCP relative to that of the control were obtained by either acidification or alkalisation followed by exhaustive dialysis against bulk milk. The spectra were obtained after the background adjustment using the corresponding supernatants obtained by ultracentrifugation.

Band Assessment	Side Chain	β-Sheet	Random Coil	α-Helix	β-Turn	Aggregated β-Sheet
Band Frequency (cm^−1^)	1608–1611	1620–1631	1640–1649	1658–1666	1668–1681	1689–1694
MCP_7_	1.1 ± 0.02 ^g^	11.8 ± 0.20 ^ab^	16.9 ± 1.34 ^a^	24.2 ± 1.88 ^abc^	44.8 ± 0.98 ^ab^	1.2 ± 0.21 ^ef^
MCP_26_	1.9 ± 0.23 ^fg^	6.6 ± 0.35 ^d^	15.0 ± 2.28 ^ab^	21.6 ± 0.49 ^bc^	52.3 ± 2.20 ^a^	2.4 ± 0.13 ^def^
MCP_31_	3.8 ± 0.11 ^e^	6.7 ± 0.03 ^d^	11.4 ± 0.59 ^ab^	22.5 ± 0.03 ^bc^	52.0 ± 0.02 ^a^	3.5 ± 0.43 ^de^
MCP_42_	2.3 ± 0.09 ^f^	1.9 ± 0.33 ^e^	13.9 ± 1.27 ^ab^	33.3 ± 1.72 ^a^	48.4 ± 2.76 ^ab^	0.1 ± 0.02 ^f^
MCP_58_	9.5 ± 0.45 ^d^	3.2 ± 0.33 ^e^	9.9 ± 0.42 ^b^	23.5 ± 1.81 ^bc^	50.6 ± 2.75 ^a^	2.9 ± 0.42 ^de^
MCP_67_	10.3 ± 0.03 ^d^	9.8 ± 0.67 ^bc^	11.9 ± 3.61 ^ab^	30.5 ± 1.09 ^ab^	34.0 ± 5.57 ^cd^	4.8 ± 0.03 ^d^
MCP_100_	15.2 ± 0.28 ^c^	6.9 ± 2.09 ^cd^	11.1 ± 1.57 ^ab^	15.1 ± 6.27 ^c^	40.8 ± 0.54 ^bc^	10.8 ± 1.80 ^c^
MCP_113_	32.6 ± 0.18 ^a^	13.2 ± 0.18 ^a^	10.1 ± 0.22 ^b^	2.6 ± 0.10 ^d^	21.9 ± 0.05 ^e^	19.5 ± 0.37 ^b^
MCP_129_	24.1 ± 0.42 ^b^	6.5 ± 0.02 ^d^	17.2 ± 0.13 ^a^	2.8 ± 0.00 ^d^	26.7 ± 0.25 ^de^	22.7 ± 0.29 ^a^

The subscripts indicate proportion of retained MCP relative to that of the control; The small letters show significant differences within the columns (*p* < 0.05); the results are expressed as the means ± standard deviation.

## Data Availability

Data are contained within this article.

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
