# Peer review of "Structural Properties of Casein Micelles with Adjusted Micellar Calcium Phosphate Content"

_foods, 2024, doi:10.3390/foods13020322_

Round 1

Reviewer 1 Report

Comments and Suggestions for Authors

In my point of view, after reviewing this manuscript, I found that the manuscript is scientifically valid and technically accurate in its methods and results. Also, the writing style or language is interesting as there is not a single grammatical error in the manuscript but there are some comments that need to change so I recommended this manuscript status is a minor revision. Some comments need to be revised as follows:

Abstract

I think the abstract needs improvement such as adding quantitative information for your results for casein size or turbidity. About abbreviations like FTIR, it is first mentioned, should write the meaning of it.

Introduction

The introduction is very simple and expressive of the subject of the study. while there are a few comments as follows:

Line 27-28: it is not clear to understand, please rewrite this sentence.

Line 44-45: I believe that more can be written in the context of the research topic through these studies.

Line 49-51: please, can you clarify this sentence by writing it again?

Materials and Methods

The Materials and Methods part is very good because it is integrated, consistent, and very well written.

 Results and discussion

-please, put figure 2 before section 3.4.

Author Response

Reviewer #1

In my point of view, after reviewing this manuscript, I found that the manuscript is scientifically valid and technically accurate in its methods and results. Also, the writing style or language is interesting as there is not a single grammatical error in the manuscript but there are some comments that need to change so I recommended this manuscript status is a minor revision. Some comments need to be revised as follows:

Abstract

I think the abstract needs improvement such as adding quantitative information for your results for casein size or turbidity. About abbreviations like FTIR, it is first mentioned, should write the meaning of it.

The abstract has been modified. 

Introduction

The introduction is very simple and expressive of the subject of the study. while there are a few comments as follows:

Line 27-28: it is not clear to understand, please rewrite this sentence.

It has been rewritten to provide more clarity. 

Line 44-45: I believe that more can be written in the context of the research topic through these studies.

It has been addressed in Line 46-52.

Line 49-51: please, can you clarify this sentence by writing it again? It has been rewritten.

Materials and Methods

The Materials and Methods part is very good because it is integrated, consistent, and very well written.

 Results and discussion

-please, put figure 2 before section 3.4.

Figure 2 has been inserted in section 3.4 

Reviewer 2 Report

Comments and Suggestions for Authors

The present study investigated the structural properties of casein micelles subjected to micellar calcium phosphate content adjustment through ICP-AES, dynamic light scattering, FTIR, 1H-NMR, etc. Overall, this manuscript is well-written, interesting, and innovative. Hence, I suggest it needs some necessary modifications as follows:

1. Materials and Methods: I suggest that the first paragraph of this part should be “2.1. Materials”, which was used to list the reagents, consumable items, and special materials utilized in this study. Also, their sources, suppliers, and/or purities should be presented.

2. Results: I strongly suggest to move Figs. S1 & S2 from the supplementary data to the main body of this manuscript, since they are very important data for FTIR spectra analysis and particle size distribution description.

3. Also Results: It is also necessary to provide the original FTIR spectra of samples with all important chemical bonds/bands marked rather that only present the amide I region through Table 3. 

4. Fig. 2: Data should be shown as means ± standard deviations.

5. Discussion: Long. I suggest to divide it into several sub-sections with individual sub-titles.

6. References: Many cited literatures are too old, even published within the last century (before the year 2000). I strongly suggest to update these literatutres to recently published peer-reviewed references.

Author Response

Reviewer #2

The present study investigated the structural properties of casein micelles subjected to micellar calcium phosphate content adjustment through ICP-AES, dynamic light scattering, FTIR, 1H-NMR, etc. Overall, this manuscript is well-written, interesting, and innovative. Hence, I suggest it needs some necessary modifications as follows:

  1. Materials and Methods: I suggest that the first paragraph of this part should be “2.1. Materials”, which was used to list the reagents, consumable items, and special materials utilized in this study. Also, their sources, suppliers, and/or purities should be presented.

This question has been addressed in Line 60-65.

  1. Results: I strongly suggest to move Figs. S1 & S2 from the supplementary data to the main body of this manuscript, since they are very important data for FTIR spectra analysis and particle size distribution description.

It has been relocated to the main body of the manuscript.

  1. Also Results: It is also necessary to provide the original FTIR spectra of samples with all important chemical bonds/bands marked rather that only present the amide I region through Table 3.

It has been added to supplementary data.

  1. Fig. 2: Data should be shown as means ± standard deviations.

It has been corrected.

  1. Discussion: Long. I suggest to divide it into several sub-sections with individual sub-titles.

Three sub-sections have been added.

  1. References: Many cited literatures are too old, even published within the last century (before the year 2000). I strongly suggest to update these literatures to recently published peer-reviewed references.

While there are some new references, including ours, the work is based on the original references and relevant references published over the years.

Please see the attached file for responses

Reviewer 3 Report

Comments and Suggestions for Authors

-        In lines 14 and 15, the abbreviations FTIR, H NMR and P NMR must be written out in full.

-        In line 54, the authors mentioned that the milk samples were pasteurized skim milk. The question is: Why were the milk samples not taken from skimmed raw milk?

-        In line 109, the samples were analysed with an FTIR spectrometer in the range of 4000–600 cm-1. The question is: is this range the right one for the analysis or can a larger range of 5000 to 850 be used?

-        In the Results section (line 143), the authors made a systematic error by discussing some results in this section, even though there is a separate section for discussion afterwards (line 279). Therefore, the authors must write the results correctly.

-        The reference list is fine, only for numbers 33 and 41 the publication date needs to be written in bold.

Author Response

Reviewer #3

In lines 14 and 15, the abbreviations FTIR, H NMR and P NMR must be written out in full.

It has been corrected.

-        In line 54, the authors mentioned that the milk samples were pasteurized skim milk. The question is: Why were the milk samples not taken from skimmed raw milk?

Because of the large volumes of milk needed in these experiments, due to the volumes required for dialysis, we needed to obtain the milk from a commercial dairy. Because of process design in the factories, raw skimmed milk cannot be taken, but pasteurized skim milk can. Due to the limited heat treatment in the pasteurization (typical HTST, 72°C for 15 sec, or equivalent) it would cause little or no change to casein micelles or whey protein denaturation, but does have the big benefit of inactivating micro-organisms and thus making the material much easier to deal with in the experimental setup. We’re convinced that using raw skimmed milk would have given similar outcomes, yet experimentally it could have been much more risky due to undesirable activities.

-        In line 109, the samples were analysed with an FTIR spectrometer in the range of 4000–600 cm-1. The question is: is this range the right one for the analysis or can a larger range of 5000 to 850 be used?

In the given context, the standard range of 4000–600 cm⁻¹ is commonly used for FTIR spectroscopy. Some examples are in the following references and our research group has always used the same range with the main focus at one particular segment – Amide I. While the range 5000-850 cm-1 can be applied in certain cases, the additional parts of this range do not provide any additional insights for milk proteins.

Kher, A., Udabage, P., McKinnon, I., McNaughton, D., & Augustin, M. A. (2007). FTIR investigation of spray-dried milk protein concentrate powders. Vibrational Spectroscopy44(2), 375-381.

Andrade, J., Pereira, C. G., de Almeida Junior, J. C., Viana, C. C. R., de Oliveira Neves, L. N., da Silva, P. H. F., ... & dos Anjos, V. D. C. (2019). FTIR-ATR determination of protein content to evaluate whey protein concentrate adulteration. Lwt99, 166-172.

Ye, M. P., Zhou, R., Shi, Y. R., Chen, H. C., & Du, Y. (2017). Effects of heating on the secondary structure of proteins in milk powders using mid-infrared spectroscopy. Journal of dairy science100(1), 89-95.

-        In the Results section (line 143), the authors made a systematic error by discussing some results in this section, even though there is a separate section for discussion afterwards (line 279). Therefore, the authors must write the results correctly.

This has been addressed and modified, and references were adjusted accordingly.

-        The reference list is fine, only for numbers 33 and 41 the publication date needs to be written in bold.

Reference 41 has been corrected and reference 33, being a book, should not be formatted in bold according to the foods journal’s guideline.

Please see the attached file for responses

Reviewer 4 Report

Comments and Suggestions for Authors

The present manuscript gives insight into the micellar structure. The results are supported by an extensive discussion and the experimental design is not only appropriate to the objective of the research but also is well described given its complexity.

In that sense, I suggest adding in Figure 1 sample fractionation and introduce the terms used in tables, e.g., “10 kDa permeat”, “Pronase-treated”, which I assume was not performed in MPCs 42, 67 and 113 since data are not available. The authors should tell what N/A stands for in Table 1 and give information on that matter in Materials & Methods section. Furthermore, MCP turbidity is compared to that of milk serum (lines 179-180) but according to Figure 1, turbidity was only measured in samples before ultracentrifugation. It needs to be clarified.

The authors state that “the Pi peak exhibited consistency” but “a broader width is observed” (lines 253-255). In fact, 31P NMR spectra of Figure 3 show that Pi peak is out of range for all samples and the broadening means that, avoiding over-range, higher peaks would have been observed. What do the authors mean by consistency? In the same way, how can the author state that changes occurred in signals H NMR spectra that are over-range (lines 272-274)? Over-range should have been avoided by loading lower amounts of sample.

Supplementary Table 2, referenced in line 175, is missing in the supplementary data document.

The authors state that “Increasing of MCP content cause slight decrease in the concentrations of individual soluble caseins” (line 194). I assume that they refer to MCPs 113 and 129, but this “slight decrease” can not be observed in Figure 2. It needs to be clarified.

I suggest:

- to add error bars in Figure 2,

- to transpose rows and columns in Table 3.

Typo errors: physicochemical (line 174), that (line 156).

Use of both UK and USA English should be avoided (e.g., analysed and analyzed).

Add coma after NMR in line 133.

Comments on the Quality of English Language

Some typographical errors. 

Author Response

Reviewer 4

The present manuscript gives insight into the micellar structure. The results are supported by an extensive discussion and the experimental design is not only appropriate to the objective of the research but also is well described given its complexity.

In that sense, I suggest adding in Figure 1 sample fractionation and introduce the terms used in tables, e.g., “10 kDa permeat”, “Pronase-treated”, which I assume was not performed in MPCs 42, 67 and 113 since data are not available. The authors should tell what N/A stands for in Table 1 and give information on that matter in Materials & Methods section.

Thank you for your suggestion, it has been modified.

Furthermore, MCP turbidity is compared to that of milk serum (lines 179-180) but according to Figure 1, turbidity was only measured in samples before ultracentrifugation. It needs to be clarified.

Clarification has been added in line 111.

The authors state that “the Pi peak exhibited consistency” but “a broader width is observed” (lines 253-255).

The Pi peak appears consistently in the expected range, we made this clarification in the text, line 270-271.

In fact, 31P NMR spectra of Figure 3 show that Pi peak is out of range for all samples and the broadening means that, avoiding over-range, higher peaks would have been observed. What do the authors mean by consistency?

Thank you for your comment, since this observation is qualitative we believe that the overloading did not have a major impact on the observed broadening. The consistency has been explained above.

In the same way, how can the author state that changes occurred in signals H NMR spectra that are over-range (lines 272-274)? Over-range should have been avoided by loading lower amounts of sample.

Thank you for your advice, however in this particular case the peaks of interest do not fall in the overloading range, and probably overloading did indeed assist us to visualise the changes better.

Supplementary Table 2, referenced in line 175, is missing in the supplementary data document.

The correction has been made, and the word "supplementary" has been removed.

The authors state that “Increasing of MCP content cause slight decrease in the concentrations of individual soluble caseins” (line 194). I assume that they refer to MCPs 113 and 129, but this “slight decrease” can not be observed in Figure 2. It needs to be clarified.

The clarification has been included in line 215. 

I suggest:

- to add error bars in Figure 2, It has been added.

- to transpose rows and columns in Table 3. The table has been transposed as per your suggestion. Thank you for your guidance; the modifications have improved clarity. 

Typo errors: physicochemical (line 174), that (line 156). It has been corrected.

Use of both UK and USA English should be avoided (e.g., analysed and analyzed). It has been corrected.

Add coma after NMR in line 133. It has been corrected.

Please see the attached file for responses
